# Effects of arthropod inquilines on growth and reproductive effort among metacommunities of the purple pitcher plant (*Sarracenia purpurea* var. *montana*)

**Rebecca E. Hale** [1]*, **Elise Powell**[1], **Leila Beikmohamadi**[1¤a], **Mara L. Alexander**[2¤b]

**1** Biology Department, The University of North Carolina Asheville, One University Heights, Asheville, North Carolina, United States of America, **2** Asheville Ecological Services Office, United States Fish and Wildlife Service, Asheville, North Carolina, United States of America

¤a Current address: Psychology Department, Western Carolina University, Cullowhee, North Carolina, United States of America
¤b Current address: United States Fish and Wildlife Service, Alexandria, Virginia, United States of America
* rhale@unca.edu

**Data Availability Statement:** All data are in a single .csv file available from the Dryad database (https://

## Abstract

Many plant species harbor communities of symbionts that release nutrients used by their host plants. However, the importance of these nutrients to plant growth and reproductive effort is not well understood. Here, we evaluate the relationship between the communities that colonize pitcher plant phytotelmata and the pitcher plants' vegetative growth and flower production to better understand the symbiotic role played by phytotelma communities. We focus on the mountain variety purple pitcher plant (*Sarracenia purpurea* var. *montana*), which occurs in small and isolated populations in Western North Carolina. We found that greater symbiont community diversity is associated with higher flower production the following season. We then examined geographic variation in communities and found that smaller plant populations supported less diverse symbiont communities. We relate our observations to patterns of community diversity predicted by community ecology theory.

## Introduction

Animal symbionts provide important services for countless plant species, including dispersing seeds [1,2], deterring herbivores [3,4], and increasing nutrient availability [5,6]. Although myriad plant species harbor symbionts that provide these benefits, studies on their effects on plant growth and reproduction are limited to the effects of single symbiont species. Yet countless plants harbor symbiont communities, which may have complex effects on hosts depending on the community's composition rather than simply its presence or absence.

Single-symbiont exclusion experiments have revealed much about the effects of symbionts on plant growth and reproduction. For example, Oliveira et al. [7] found that excluding ant symbionts resulted in eight times more herbivory scars from moth larvae in prickly pear, *Opuntia stricta*, and when ants were present, plants had 50% higher fruit production. Horvitz

datadryad.org/stash/share/9Z3gogbGOlzisvv1uypq
bG2ooz05TXM2keJw9scMcEc).

**Funding:** This project was funded by grants from the UNC Asheville Undergraduate Research Program to E. Powell and L. Beikmohamadi, a grant from the University of North Carolina Asheville University Research Council to R. Hale, J. Rhode Ward, and Caroline Kennedy, and a grant from the U.S. Fish and Wildlife Service, Ecological Services Asheville Field Office to R. Hale, J. Rhode Ward, and C. Kennedy.

**Competing interests:** The authors have declared that no competing interests exist.

and Schemske [8] found that excluding symbiotic ants from *Calathea ovandensis* reduced seed set by 50%, mediating the effect of an herbivorous moth larva. In a meta-analysis of similar ant-exclusion studies, Trager et al. [4] found that symbiotic ants reduce herbivory and increase reproductive output (i.e., production of flowers, fruit or seeds) by an average of 49%.

Whereas these studies clearly show animal symbionts benefit their plant hosts, they describe the relationship between the plant and a single symbiont. However, many plant species harbor rich communities of symbionts that vary widely in composition among hosts. In nutrient-poor habitats, animal symbionts provide their plant hosts with organic nutrients in the form of prey carcasses, feces, and their bodies [6,9,10]. Large symbiont communities are common in pitcher plants and bromeliads that typically grow in nutrient-poor habitats and that form phy-totelmata–pools of water formed by a plant's leaves. For example, *Nepenthes* sp. pitcher plants can harbor 59 species of inquilines (aquatic symbionts); *Sarracenia purpurea* pitchers house as many as 165 species [11]. Seventy species have been described from the phytotelmata of *Vriesea* sp. bromeliads [12].

Data on the effects of nutrient-releasing symbionts on vegetative growth are rare. Romero et al. [6] used stable isotope analyses to show that prey carcasses and feces from symbiotic jumping spiders increased nitrogen availability for *Bromelia balansae*, and that plants with spiders had more and longer leaves. In a similar study of the effects of a symbiotic treefrog on nutrient uptake in *Vriesea bituminosa*, Romero et al. [9] found that net photosynthetic rate and transpiration rate were both approximately 30% higher in plants that received frog feces or prey debris than in those that did not, though the effect varied seasonally.

In plant species hosting large symbiont communities, the absence of a single species may be less important than the community's diversity or the total number of organisms in the community. Further, variation in community diversity or organism abundance may be influenced by the size of host plant population or its connectivity to other populations. Classic studies of island biogeography [13] and more recent research on habitat fragmentation and metacommunity dynamics [14] suggest that larger plant populations should harbor more diverse symbiont communities if the plant population functions as the patch, as larger islands [15] and larger patches within metacommunities [16–18] typically harbor more species. Alternatively, if the individual plant functions as the patch, a larger network of patches (i.e., a larger plant population) could harbor more species if it provides a greater diversity of niches to occupy [19]. Isolation of host plant populations are likely an important factor affecting symbiont community diversity [20], though its effect on species diversity varies widely. Species richness can increase [21,22], decrease [21], or not change [17,23] with patch isolation. However, isolation should have a strong effect on symbiont dispersal, and dispersal among patches strongly influences community diversity in many systems [24–26].

The inquiline communities that inhabit the leaves of the purple pitcher plant, *Sarracenia purpurea*, long have been models for the study of community diversity and metacommunity dynamics [e.g., 27–32]. The purple pitcher plant inhabits bogs and other nutrient-limited habitats and supplements its nitrogen and phosphorus intake by digesting prey, typically Dipteran and Hymenopteran species [33], trapped within its pitcher-shaped leaves. These prey are processed by and support a diverse community consisting of bacteria, rotifers, mosquitoes, midges, mites, and other invertebrates [34–37]. Larvae of the pitcher plant midge, *Metriocnemus knabi*, shred prey carcasses, releasing particles that feed bacteria, protozoa, rotifers, and copepods [36]. These organisms are consumed by mites and mosquito larvae, most commonly of the pitcher plant mosquito, *Wyeomyia smithii* [38]; thus, midges and mosquitoes have a commensal relationship, with both depending on prey abundance [35]. A diverse inquiline community is associated with higher available nitrogen (ammonia) in pitcher fluid, which is taken up by plant tissue [39]. Given the nutrient limitations of the bog environments where

these pitcher plants grow, relationships among symbionts likely represent a critical factor affecting the growth and reproductive effort of the plants.

In the mountains of western North Carolina, a rare variety of the purple pitcher plant, *S. purpurea* var. *montana* (Southern Appalachian purple pitcher plant) [40], inhabits mountain bogs that are geographically isolated. Small and/or isolated plant populations may not be able to support some symbiont species due to their limited vagility. Previous studies have found that community structure can vary significantly among individual pitchers, plants, and plant populations [27,37,41]. The consequences of reduced diversity or shifted composition of the inquiline community could have cascading effects on the plants themselves, such as reduced growth or reproductive effort.

This study examines the relationships between inquiline community composition and host plant growth, reproductive effort, population size, and population isolation. Specifically, we examined eight *S. purpurea* var. *montana* populations occurring across four counties in western North Carolina to address two objectives. The first objective was to determine whether there exists a relationship between inquiline community composition and vegetative growth or reproductive effort. We tested the hypothesis that seasonal vegetative growth would increase with the (H1a) number of organisms in pitchers and with (H1b) the diversity of organisms in pitcher communities. We also hypothesized that rosettes that flower will have had (H2a) more organisms and (H2b) greater diversity of organisms in its pitchers than rosettes that do not flower.

The second objective was to determine whether a relationship exists between inquiline community composition and host plant population size (i.e., the number of colonizable plants) or isolation, as more host plants and greater connectivity to other plant populations may facilitate pitcher colonization and result in larger inquiline populations. We hypothesized that a larger host plant population size would be associated with more organisms in pitchers (H3a) and greater inquiline diversity (H3b). We further hypothesized that the distance to the nearest neighboring plant population would be negatively associated with the number of organisms in pitchers (H4a) and the diversity of organisms in pitcher communities (H4b).

## Materials and methods

### Study sites

We sampled eight *S. purpurea* var. *montana* populations occurring across four counties in western North Carolina (Table 1; range: 35.0–35.3N, 82.5–83.2W, 750–1150 m elevation) under North Carolina Protected Plant Permit No. 456, issued by the North Carolina Department of Agriculture and Consumer Services, Plant Conservation Program. Access to sites was granted by The Nature Conservancy, the U.S. Forest Service, Highlands Biological Station, the North Carolina Plant Conservation Program, the Highlands-Cashiers Land Trust, and a private land owner. Specific localities for the sites are not provided at the request of landowners and the U.S. Fish and Wildlife Service, out of concern that plants at some sites may be poached.

Data regarding *S. purpurea* var. *montana* population sizes were obtained during 2012 and 2013 by the U.S. Fish and Wildlife Service (M. Alexander, unpublished data) by counting the number of distinct clumps of pitchers. Although clumps may include multiple, genetically distinct plants, clumps were defined operationally to be groups of pitchers extending from a single meristem or multiple overlapping meristems, and physically separated from other groups of pitchers by 15 cm. We assume that the number of clumps is correlated with the total number of pitchers at the site, where the number of pitchers at a site is the ecologically relevant metric for the number of colonizable patches within a population. We transformed these

**Table 1. Population size, distance to nearest neighboring population, and location data for eight *s. purpurea* var. *montana* populations in Western North Carolina.**

| Population | Clumps[a] | Rank size | NND (m) | Rank distance | N[b] | County | Elevation[c] (m) | Approx. mean rainfall[d] (cm) 2015 | 2016 |
|---|---|---|---|---|---|---|---|---|---|
| A | 7 | 1 | 14,119 | 8 | 30 | Henderson | 634 | 155 | 110 |
| B | 33 | 2 | 3,138 | 3 | 60 | Transylvania | 820 | 180 | 130 |
| C | 50 | 3 | 3,000 | 5.5 | 60 | Macon | 1171 | 160 | 130 |
| D | 100 | 4 | 451 | 1.5 | 60 | Transylvania | 870 | 210 | 130 |
| E | 105 | 5 | 3,000 | 5.5 | 40 | Macon | 996 | 180 | 130 |
| F | 124 | 6 | 451 | 1.5 | 60 | Transylvania | 848 | 220 | 130 |
| G | 297 | 7 | 8,664 | 7 | 40 | Jackson | 927 | 220 | 130 |
| H | 300 | 8 | 3,100 | 4 | 30 | Transylvania | 892 | 220 | 110 |

[a] Population size (measured as number of pitcher clumps) and distance to the nearest neighboring population (NND) were converted to rank population size (1 = smallest) and rank nearest neighbor distance (1 = shortest distance), respectively, for analyses. Where ranks were tied, ranks were averaged and assigned to both populations.

[b] Number of pitchers sampled over three months.

[c] Data from Google Earth Pro version 7.3.2 (©2018 Google, Inc.).

[d] Data by county from the U.S. National Weather Service [42].

estimates of population size to population size rank for analyses, with the smallest site assigned a rank of 1 (Table 1). Pairwise geographic distances among all eight populations were estimated using Google Earth Pro version 7.3.2 (©2018 Google, Inc.) and the nearest neighbor distances were determined from these values. Specifically, nearest neighbor distance was simply the distance to the nearest identified population. In some cases, the nearest population was not part of our study. Distances were also transformed into ranks, with the sites with the shortest neighbor distance assigned a rank of 1.

## Inquiline communities

At six of the eight sites, twenty *S. purpurea* var. *montana* rosettes were selected for sampling. At H, fifteen rosettes were sampled. A meter tape was laid along the margin of the population to form a transect and focal rosettes were evenly spaced along the transect. Rosettes were defined as a single meristem to which a cluster of pitchers and/or pitcher buds could be traced. These rosettes likely were connected to other rosettes via rhizomes, but their connectivity could not easily be traced without causing damage. To reduce the likelihood of sampling rosettes from the same genet, we did not sample rosettes within a 0.5 m radius of one another and avoided sampling rosettes from the same clump. However, at A, only two large clumps were present; therefore, we sampled five rosettes in each of the two clumps for a total of 10 rosettes. Within these constraints, rosettes were selected based on the presence of new pitchers during the first sampling day. Sampled pitchers were marked with colored beads strung on a monofilament necklace.

Sampling was repeated for the months of June, July, and August in 2015. Sampling consisted of pipetting the entire contents of the pitcher into a 50 ml microcentrifuge tube. The fluid volume of each sample was recorded and samples were returned to the lab and stored at 4 ˚C. We also measured the total volume of the sampled pitcher by filling the emptied pitcher to the lip with tap water aged to allow evaporation of chlorine, and then quantifying the water volume as it was removed with a pipette.

To capture inquiline communities at approximately the same age and stage of succession, young pitchers were selected. During previous work in this region, we had observed new pitcher growth in May. Young pitchers have a softer texture and are paler green than older pitchers. Further, pitchers that have overwintered commonly have browned margins and are easily distinguished early in the season from new growth. Therefore, in June, only young pitchers that were perceived to have not overwintered were sampled, and their communities were assumed to be less than one month old. After the first pitcher was sampled on each rosette, an unopened pitcher on the same rosette was marked for sampling the following month. Marking of new, unopened pitchers ensured that at the next sampling date, the community was less than one month old, allowing us to control for community age while comparing communities from different months. The sampling procedure was repeated in July and August.

On the University of North Carolina Asheville campus, samples were filtered using 100 μm Nitex mesh (Sefar Inc., Buffalo, NY) to retrieve organisms and debris from the samples. The material collected in the filter was preserved in 70% ethanol. Arthropods present in the preserved samples were quantified under a dissecting microscope. Dipteran species observed in the samples included the mosquito *Wyeomyia smithii* and the midge *Metriocnemus knabi*. We also observed a variety of mite morphotypes that likely include the mite *Sarraceniopus gibsoni* but are yet unidentified. These mites are combined in a single group for analyses. We occasionally observed copepods, but smaller organisms, including rotifers and unicellular eukaryotes, were not quantified.

## Vegetative growth and reproductive effort

Reproductive effort was quantified as the number of flowers stemming from marked rosettes. To measure vegetative growth during a given year, we counted all pitchers without apparent winter browning that were emerging from the meristem of focal rosettes. In 2015, flowers were counted in July and sites were revisited in October and November 2015 to count pitchers on marked rosettes. We returned to each site in August 2016 and quantified total flower stalks and non-overwintered pitchers on the same rosettes.

## Data analysis

All analyses were conducted in R version 3.5.0 (R Development Core Team 2018) except where noted. We calculated a Spearman's rank correlation matrix on the abundance of midges, mosquitoes, copepods, and total mites using the rcorr() function of the package Hmisc v. 4.1–1 [43]. Shannon's Diversity Index (H) was calculated using the diversity() function of the vegan package v. 2.5–4 [44].

We evaluated whether inquiline communities influence vegetative growth (H1) with generalized linear models using the glm() function. In all models, number of pitchers was the dependent variable and the independent variables included one of the two community measurements (a: total organisms and b: Shannon's H), site, and log(fluid volume). Also, we used generalized linear models to test whether the two community measurements differed between rosettes that flowered and those that did not flower (H2a-b), including site and log(fluid volume) as additional independent variables. In all analyses, community and fluid volume measurements from pitchers on the same rosette were averaged, thereby precluding month as a variable in the models. To examine whether the effect of community is delayed, we evaluated statistical models with 2015 community measures and with both 2015 and 2016 measures of pitcher count and whether rosettes flowered. A model-comparison approach was used to evaluate the relative fit of models with progressively fewer fixed effects, starting with models including all interaction terms. Model fit was evaluated using AIC [45], by identifying

the model with the lowest AIC values. Marginal effects were estimated using the emmeans() function of the emmeans package v. 1.4.5 [46]. R2 values were calculated using the rsq(type = 'v') function in the rsq package of R [47].

We used Spearman's rank correlations to determine if host plant population size (H3a-b) or population distance from the nearest neighboring population (H4a-b) correlated with number of organisms in pitchers and Shannon's H across sites using the cor.test() function. To obtain site averages for inquiline abundance and diversity, we calculated mean number of organisms per pitcher and mean Shannon's H per pitcher by calculating the means of the rosette averages (as opposed to calculating the mean of all pitchers across rosettes).

## Results

### Inquiline communities

A total of 380 pitchers were sampled over three months in 2015. Sites E and G were not sampled in June. All eight populations were sampled in July, and site H was not sampled in August. Invertebrate colonists included larvae of one species of mosquito (*Wyeomyia smithii*) and one species of midge (*Metriocnemus knabi*), as well as a large number of mites that are combined as a single taxon in our analyses. These mites likely consisted of nymph and adult stages of *Sarraceniopus gibsoni*. Additionally, copepods, bdelloid rotifers, and protozoa (all unidentified) were occasionally observed. We did not observe Cladocerans in any samples, though they are known to occur in this species [27].

Taxon richness varied among pitchers from 0–4 and total abundance of organisms ranged from 0–364 individuals. Although taxon richness was relatively low, in part due to our level of taxon identification, richness varied considerably. No mosquitoes were found in site A (N = 30), which is both the smallest plant population and the most isolated site, nor in site B (N = 60). Copepods were missing from sites A (N = 30), B (N = 60), C (N = 60), and F (N = 60). All populations had midge larvae and mites. When all sites and months were pooled, the abundance of mites was positively correlated with the abundance of mosquitoes and midges (Table 2).

### Vegetative growth and reproductive effort

Community variables had varying effects on the number of pitchers produced per rosette, with no consistent, directional support for hypothesis 1. The number of organisms in pitchers had a significant influence on the number of pitchers produced in 2015, but its effect differed among sites and covaried with the volume of water in pitchers (model *c*, Table 3). Because of these interactions, the positive marginal effect of the number of organisms in pitchers on the number of pitchers (log(pitchers) = 1.97·log(no. organisms); Fig 1A) is difficult to interpret. There was no effect of the number of organisms on pitchers produced in 2016 (model *g*, Table 3; Fig 1B). Community diversity (Shannon's H) did not significantly affect the number of pitchers produced in 2015 (models *l*, *n*, and *o* were equivocal, Table 3; Fig 1C). In 2016, the effect of diversity on number of pitchers varied with fluid volume (interaction in model *k*; Table 3), such that the positive marginal effect of diversity (log(pitchers) = 1.96·H; Fig 1D) cannot be interpreted easily. Further, the amount of total variance explained by model *k* was neglible.

Support for hypothesis 2 that communities would affect flowering were also mixed. In 2015, the average number of organisms in pitchers did not differ between plants that flowered and did not flower (models *c* and *d* were equivocal, Table 4; Fig 2A), but rosettes that flowered in 2016 had, on average, 18 more organisms than rosettes that did not flower in 2016 (model *d*, Table 4; Fig 2B; estimated marginal means ± SE from the model log(organisms+1) = flowered +log(volume)+site, flowered: $e^{3.88 \pm 0.10}$ organisms, did not flower: $e^{3.42 \pm 0.07}$ organisms). In

**Table 2. Spearman rank correlation matrix relating abundances of four invertebrate taxa within pitchers.**

| Taxon | Copepods | Midges | Mites | Mosquitoes |
|---|---|---|---|---|
| Copepods | | 0.06 | -0.01 | 0.08 |
| Midges | 0.27 | | 0.19 | 0.24 |
| Mites | 0.92 | 0.0002 | | 0.09 |
| Mosquitoes | 0.14 | <0.0001 | 0.073 | |

Values above the diagonal are correlation coefficients, values below the diagonal are asymptotic probabilities. N = 380 pitchers pooled across all sites and all months.

addition, mean number of organisms increased with pitcher fluid volume and varied between sites. In neither 2015 nor 2016 did community diversity (Shannon's H) differ between flowering and non-flowering rosettes; models *i* and *j* were equivocal in both years.

Number of flowers and number of pitchers varied considerably within and among sites, as well as between years (Table 5). Qualitatively, rosettes at all but one site were less likely to flower in 2016 and produced fewer flowers on average. This region of Western North Carolina is typically very wet in the *Sarracenia* growing season and 2016 was considerably drier than

**Table 3. Comparison of model fit examining H1 that inquiline abundance and diversity affect vegetative growth.**

| | | 2015 | 2016 |
|---|---|---|---|
| **Model: log(no. pitchers) =** | | **AIC (R$^2$)$^a$** | **AIC** |
| H1a: Number of inquiline organisms | | | |
| *a* | log(no. organisms) + log(volume$^b$) + site + all 2- and 3-way interactions | 738.68 | 612.60 |
| *b* | log(no. organisms) + log(volume) + site + all 2-way interactions | 732.16 | 607.99 |
| *c* | log(no. organisms) + log(volume) + site + log(volume) x log(no. organisms) + site x log(no. organisms) | **722.96 (0.36)** | 612.67 |
| *d* | log(no. organisms) + log(volume) + site + site x log(no. organisms) | 727.83 | 612.57 |
| *e* | log(no. organisms) + log(volume) + site | 725.38 | 613.68 |
| *f* | log(volume) + site | 728.37 | 611.68 |
| *g* | log(volume) | 727.38 | **603.90 (0.0006)** |
| H1b: Shannon's H | | | |
| *h* | H + log(volume) + site + all 2- and 3-way interactions | 801.10 | 644.52 |
| *i* | H + log(volume) + site + all 2-way interactions | 739.19 | 622.11 |
| *j* | H + log(volume) + site + log(volume) x site + log(volume) x H | 730.60 | 613.51 |
| *k* | H + log(volume) + H x log(volume) | 819.42 | **602.39 (0.002)** |
| *l* | H + log(volume) + site | **729.66 (0.32)** | 613.18 |
| *m* | H + log(volume) | 803.74 | 605.64 |
| *n* | log(volume) + site | **728.37 (0.32)** | 611.68 |
| *o* | site | **727.92 (0.31)** | 609.76 |
| *p* | log(volume) | 812.99 | 604.27 |

To test the hypothesis, we evaluated whether mean number of organisms or Shannon's H affected the number of pitchers produced in 2015 or 2016. Number of organisms, Shannon's H, and fluid volume were averaged across three pitchers on each of 20 rosettes per site in 2015. Terms that did not improve fit were removed sequentially until the model with lowest AIC was identified. As a result, not all combinations of all terms are presented. Models explaining the most variance in community pitcher number are indicated in bold for each year; multiple models in bold where AIC values differed by less than 2.0.

$^a$ Lower values of AIC indicate a better fitting model. R$^2$ values are provided for models with lowest AIC.
$^b$ 'volume' indicates the amount of fluid in pitchers.

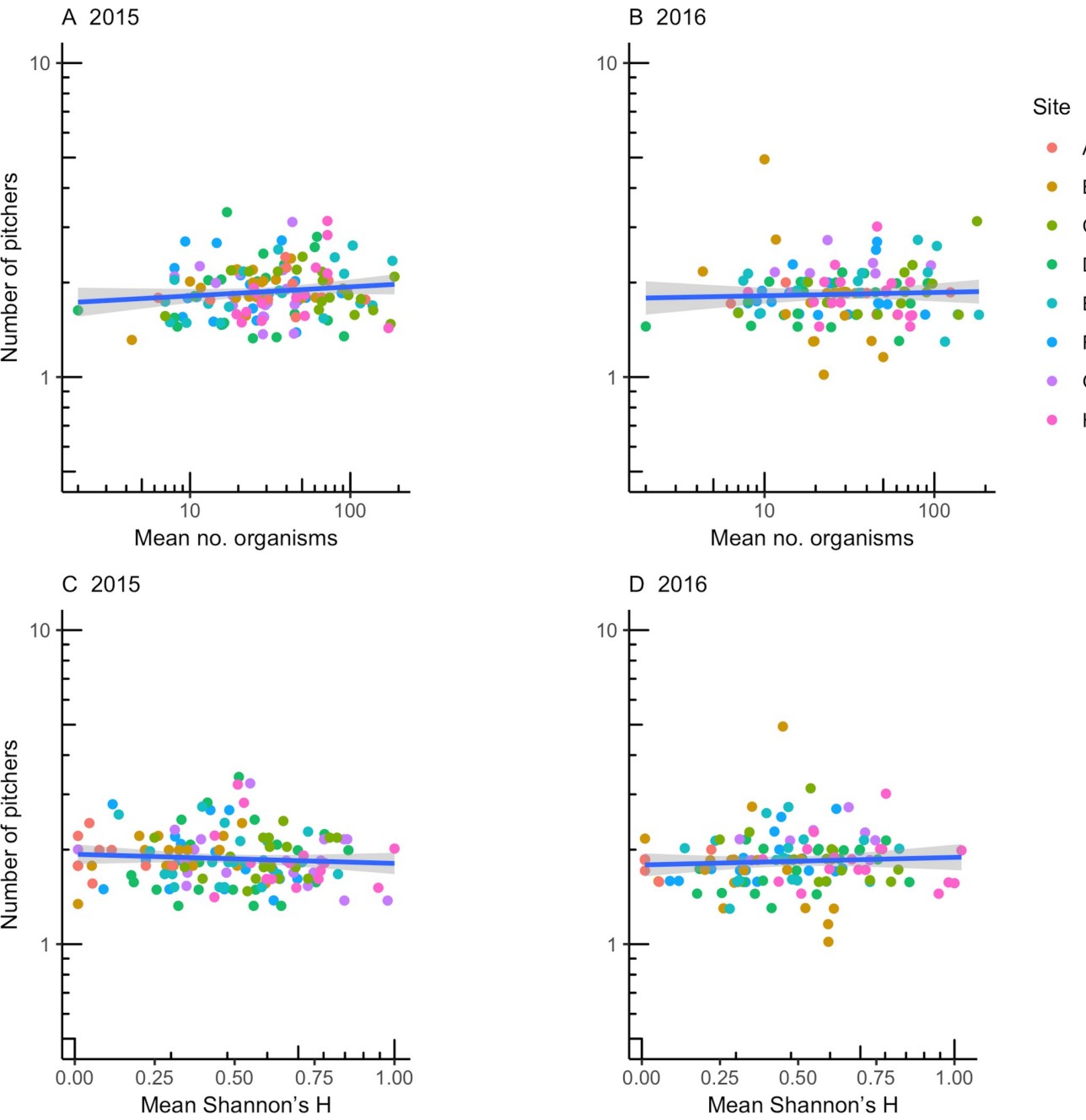

**Fig 1. Weak effects of inquiline abundance and community diversity on vegetative growth provide limited support for H1.** Number of inquiline organisms (A, B) and community diversity (Shannon's H; C, D) and their effect on number of pitchers produced in 2015 and 2016. Total organisms and Shannon's H were measured in 2015. Points are the grand mean number of pitchers plus residuals from models in Table 3. For A and B, residuals were obtained from models *f* and *g*, respectively. For C and D, residuals were obtained from models *o* and *p*, respectively.

2015 (Table 1), which may explain reduced reproductive effort. Although older or larger plants might be predicted to produce both more pitchers and more flowers, the two were not consistently correlated: pitcher count and flower count were correlated in 2016 (Pearson product moment correlation: t = 2.06, df = 120, p = 0.042, $R^2$ = 0.03), but not in 2015 (t = 0.30, df = 133, p = 0.76, $R^2$ = 0.0009).

**Table 4. Comparison of model fit examining H2 that inquiline abundance and diversity affect reproductive effort.**

| | | 2015 | 2016 |
|---|---|---|---|
| **Model** | | **AIC (R²)[a]** | **AIC** |
| H2a: log(mean no. organisms + 1) = | | | |
| a | flowered[b] + log(volume[c]) + site + log(fluid volume) x site | 297.13 | 272.94 |
| b | log(volume) + site + log(volume) x site | 296.90 | 285.74 |
| c | log(volume) + site | **289.74 (0.38)** | 281.53 |
| d | flowered + log(volume) + site | **289.98 (0.37)** | **267.34 (0.43)** |
| e | log(volume) | 321.06 | 318.84 |
| f | site | 326.08 | 304.12 |
| H2b: mean Shannon's H = | | | |
| g | flowered + log(volume) + site + log(volume) x site | -66.83 | -65.50 |
| h | log(volume) + site + log(volume) x site | -67.83 | -67.22 |
| i | log(volume) + site | **-71.36 (0.35)** | **-70.77 (0.38)** |
| j | flowered + log(volume) + site | **-70.94 (0.35)** | **-69.21 (0.35)** |
| k | log(volume) | -30.93 | -1.65 |
| l | site | -59.18 | -62.30 |

To test the hypothesis, mean number of organisms in a rosette's pitchers in 2015 was compared between rosettes that flowered and those that did not flower in 2015 and in 2016. Number of organisms, Shannon's H, and fluid volume were averaged across three pitchers on each of 20 rosettes per site in 2015. Terms that did not improve fit were removed sequentially until the model with lowest AIC was identified. As a result, not all combinations of all terms are presented. Models explaining the most variance in community measure are indicated in bold for each year; multiple models in bold where AIC values differed by less than 2.0.

[a] Lower values of AIC indicate a better fitting model. R² values are provided for models with lowest AIC.

[b] 'flowered' is a binary variable that takes values of 0 (no) or 1 (yes).

[c] 'volume' indicates the amount of fluid in pitchers.

## Population size and isolation

In testing hypothesis 3 that larger pitcher plant populations would support more organisms in pitchers and more diverse communities, we found no correlation between the rank population size and the mean number of organisms in pitchers (H3a: $\rho = 0.07$, S = 90, N = 8, p = 0.88; Fig 3A), but mean Shannon's H increased with population size (H3b: $\rho = 0.76$, S = 20, N = 8, p = 0.037; Fig 3B). In testing hypothesis 4 that more isolated populations would support less diverse communities and fewer organisms in pitchers, we found no correlation between rank distance to the nearest population and either mean number of organisms in pitchers (H4a: $\rho = 0.12$, S = 73.88, N = 8, p = 0.78) or Shannon's H (H4b: $\rho = -0.14$, S = 96, N = 8, p = 0.73).

We also calculated Shannon's H at the site level, summing individual taxon abundance across pitchers first, then calculating Shannon's H. To control for sampling effort, we analyzed data for July, only, when all sites were sampled. We found no correlation between site-level Shannon's H and the rank number of plants ($\rho = 0.17$, S = 70, N = 8, p = 0.70) or rank distance to the nearest population ($\rho = -0.18$, S = 99, N = 8, p = 0.67).

## Discussion

We expected to see evidence that plants benefit from diverse symbiont communities through greater flower production or plant growth. We found some evidence in support of our

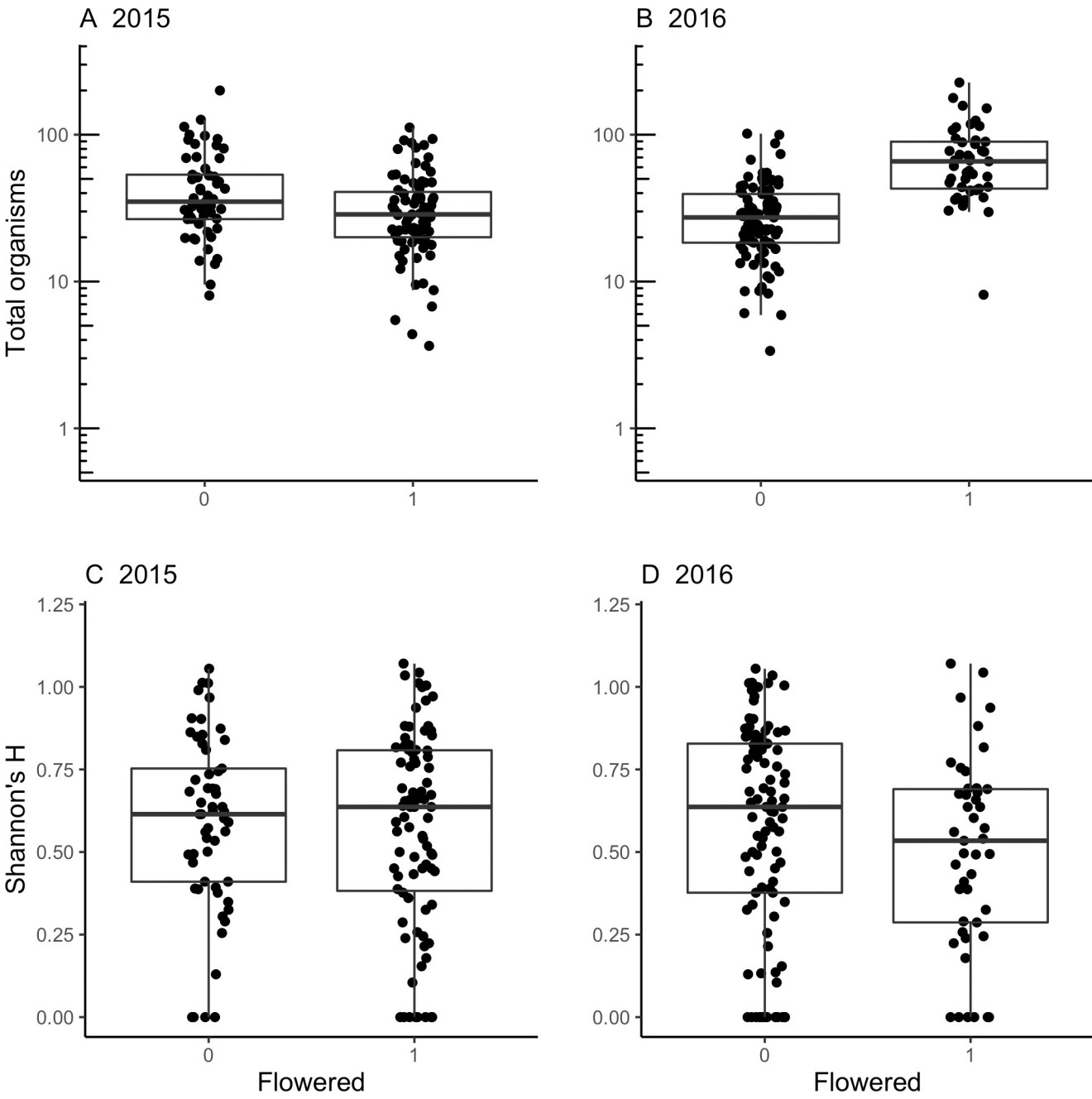

**Fig 2. Comparison of inquiline abundance and community diversity between flowering and non-flowering rosettes provides partial support for H2.**
Number of inquiline organisms (A, B) and community diversity (Shannon's H; C, D) within pitchers that flowered and those that did not flower in 2015 and 2016. Number of organisms was significantly greater among rosettes that flowering in 2016 than among those that did not flower. Number of organisms and Shannon's H were measured in 2015. Boxplots and points are least square means-adjusted values. For A and B, means for flowering and not flowering were estimated using model *d* in Table 4 and residuals were calculated from model *c*. For C and D, means for flowering and not flowering were estimated using model *j* in Table 4 and residuals were calculated from model *i*.

hypothesis. First, we found weak effects of number of organisms and Shannon's H on the number of pitchers produced in 2015. One interpretation of these effects is that they are too small to be biologically meaningful. Another interpretation is that the effects of communities are meaningful, but given the relatively few taxa we quantified (maximum four arthropod

**Table 5. Proportion of rosettes that had flowers, Mean ± SD number of flowers, and Mean ± SD number of pitchers produced in 2015 and 2016.**

| Site | 2015 | | | 2016 | | |
|------|---------------------|-------------------|-------------------|---------------------|-------------------|-------------------|
|      | Proportion flowering | Mean ± SD flowers | Mean ± SD pitchers | Proportion flowering | Mean ± SD flowers | Mean ± SD pitchers |
| A | NA[a] | NA | 4.6 ± 1.13 | NA | NA | 6.6 ± 1.06 |
| B | 0.23 | 0.27 ± 0.52 | 4.6 ± 1.12 | 0.20 | 0.20 ± 0.40 | 6.9 ± 5.74 |
| C | 0.61 | 1.12 ± 1.15 | 7.1 ± 1.95 | 0.29 | 0.38 ± 0.61 | 7.43 ± 2.86 |
| D | 0.48 | 0.65 ± 0.87 | 12.0 ± 6.61 | 0.30 | 0.3 ± 0.46 | 6.25 ± 1.86 |
| E | 0.20 | 0.20 ± 0.41 | 6.6 ± 2.36 | 0.25 | 0.25 ± 0.44 | 7.38 ± 2.66 |
| F | 0.90 | 3.00 ± 2.15 | 8.5 ± 3.38 | 0.60 | 0.80 ± 0.76 | 7.0 ± 2.23 |
| G | 0.80 | 1.32 ± 1.23 | 6.4 ± 2.63 | 0.40 | 0.37 ± 0.49 | 8.8 ± 1.88 |
| H | 0.74 | 0.97 ± 0.81 | 10.2 ± 5.09 | 0.17 | 0.17 ± 0.38 | 6.8 ± 2.31 |

[a] Site A clumps were so dense that identifying rosettes from which flowers were growing and counting the number of pitchers was not possible.

taxa), we were not able to adequately quantify them. We did not identify unicellular eukaryotes or prokaryotes, but these organisms may play important roles in nutrient cycling within the pitchers. Despite not quantifying these smaller organisms, we did detect effects of communities on vegetative growth, which may have been possible because the four arthropods have relatively large and non-overlapping trophic roles [36,38]. Bradshaw and Creelman [39] showed that when midges are present, they quickly produce a spike in available nitrogen in pitcher fluid. They found that mosquitoes have a similar, but delayed effect. In combination, midges and mosquitoes may sustain nitrogen levels that vary little over time and facilitate continued pitcher production over the season.

Another explanation for a relationship between inquiline communities and number of pitchers is that more pitchers on a plant may facilitate the maintenance of high species richness. Indeed, Buckley et al. [41] found this pattern in *S. purpurea* and attributed it to plants with more pitchers offering more area for organisms to colonize–i.e., the common positive relationship between species richness and habitat area. If this were true in our study, then we would expect species richness to increase with the number of pitchers present on the day of sampling. However, we quantified the number of pitchers at the end of the season and counted pitchers solely deemed to have been produced that year. Because *S. purpurea* pitchers that have overwintered can hold water, and because plants continue to produce new pitchers over the course of the growing season, the number of pitchers available for colonization on the day of community sampling was necessarily different than the number of annual pitchers counted at the end of the year. Therefore, a greater habitat area of plants with more pitchers does not immediately explain our result.

Second, we found that rosettes with more organisms in their pitchers in 2015 were more likely to flower in 2016. The delayed effect on flower production can be explained by the phenology of this species. Purple pitcher plants produce their first and largest pulse of flower production early in the season, before communities are established, but produce new pitchers throughout the growing season. Therefore, it is not surprising we found no relation between communities and flowering in 2015. We found an effect of total organisms, but not community diversity, on flowering in 2016. With a delayed effect such as this, the number of organisms, rather than the number of species, may determine the amount of nutrients available for flower production the following season. Whereas a diverse community may lead to a nutrient supply that varies less over time and affects immediate growth allocation, a greater number of organisms may release more nutrients overall, allowing greater nutrient storage for use in reproductive allocation the following year.

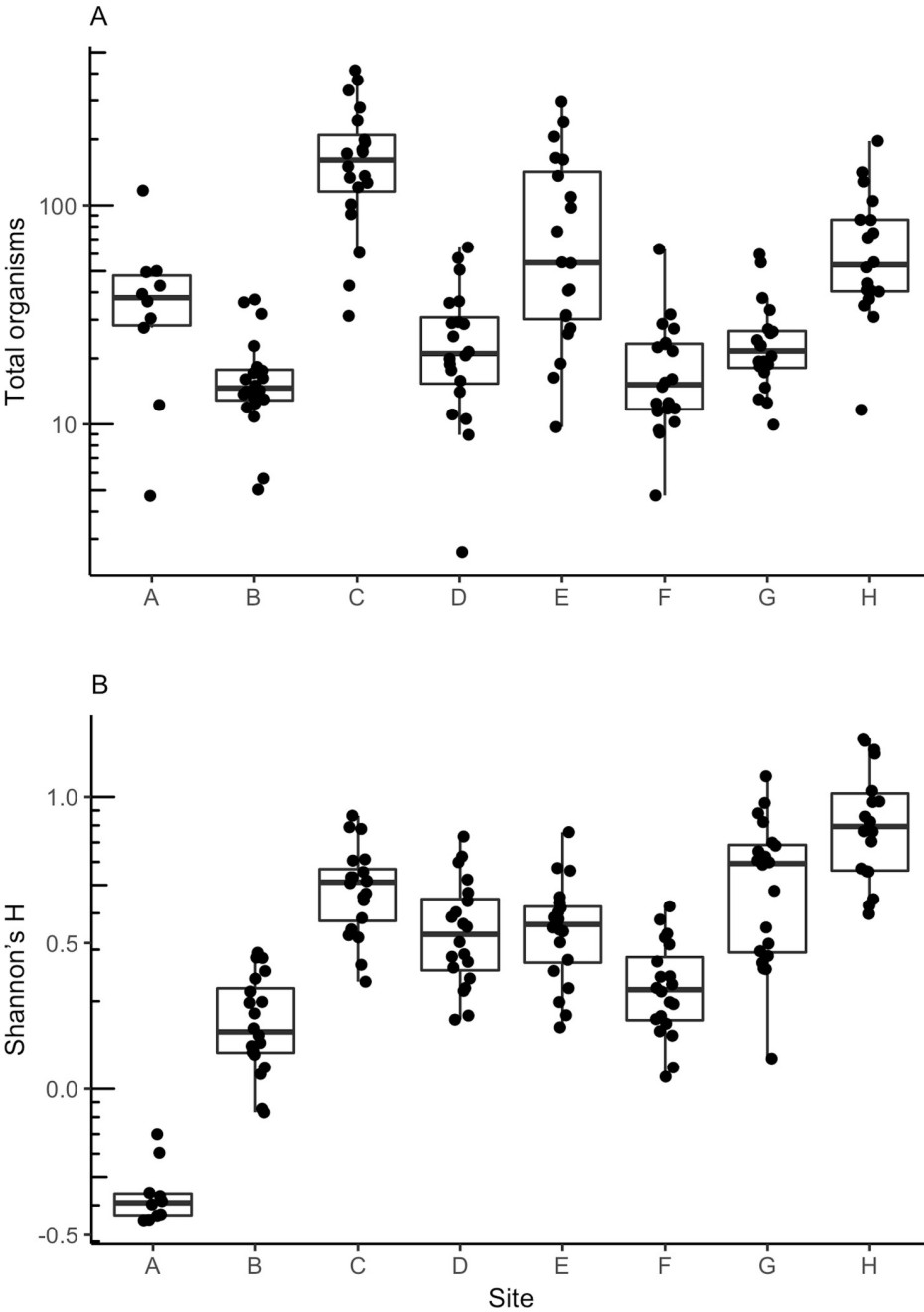

**Fig 3. Communities within pitchers varied among plant populations.** A) Total inquiline organisms and B) Shannon's H for each of eight sites sampled in July 2015. Populations are ordered from fewest plants (site A) to most plants (site H) and rank order of isolation is provided in Table 1. Boxplots and points are least square means-adjusted values. Site means were obtained for A and B from models $c$ and $i$, respectively, and residuals were obtained from models $e$ and $k$, respectively, in Table 4.

If phytotelma communities do, indeed, enhance growth and reproductive effort of rosettes, then plants in populations with smaller or less diverse communities may exhibit less growth or flower less frequently. Drawing from the community ecology literature [e.g., 13], we predicted that less diverse communities might occur where there is less area for organisms to colonize

(i.e., smaller plant populations) and where dispersal distances limit organisms from colonizing (i.e., more isolated plant populations). Consistent with the common pattern of species diversity increasing with patch or island size [e.g., 15–17] and with similar work in the bromeliad *Vriesea* sp. [48] and in *S. purpurea* [30,41], we found that the number of organisms and community diversity increased with the amount of fluid in a pitcher. We also found that community diversity, but not number of organisms in pitchers, increased with population size. Therefore, our data provide support for the hypothesis that a population of pitchers functions as one large patch that supports greater species diversity with increasing area (or volume, in this case). These results are consistent with other studies of *S. purpurea*. As described earlier, Buckley et al. [41] found that plants with more pitchers had higher species richness at the plant level. Similarly, we quantified diversity at the site level compared to number of plants in the population and found a positive correlation.

Further, we found no relationship between population isolation and either inquiline abundance or species diversity, which suggests that long-distance dispersal plays a minimal role in maintaining populations of the inquiline species in this system. The effects of isolation and dispersal distance on community diversity vary widely across systems and our results are consistent with studies that have found they have no effect [17,23,25]. Kneitel et al. [24] examined dispersal and community diversity in *S. purpurea* and found that intermediate dispersal frequencies resulted in the greatest protozoan diversity. At low frequencies, they argued that diversity is low due to high local extinction rates, whereas at high dispersal frequencies, competitive interactions among taxa reduce diversity. These two processes are balanced at intermediate frequencies, allowing coexistence of more taxa. However, when the pitcher plant mosquito, a protozoa predator, was introduced, there were no effects of dispersal frequency on diversity, and diversity was consistently lower. Our focus on predatory and scavenging species, and exclusion of protozoa, may help explain why we found no effect of population isolation on mean diversity across sites.

The absence of an effect of isolation on diversity suggests that immigration from other sites is negligible. If so, then diversity of inquiline communities may be maintained by entirely within-population processes, with populations containing more pitchers better able to maintain diverse inquiline communities. Pitchers serve as overwintering sites for dormant mosquito larvae [49] and may be refugia for other pitcher inhabitants. Therefore, we might expect the number of plants to correlate with greater abundance of at least some taxa. Although we did not analyze the relationships between plant population size or isolation and the abundances of individual community members, it is noteworthy that mosquitoes were absent from the site with the smallest plant population. It may not be coincidental that mosquitoes colonize pitchers when adult females oviposit in the pitcher water; dispersal among pitchers within populations may be an important factor explaining variation in local abundances across taxa [20,50].

Our data provide evidence that phytotelma communities benefit host pitcher plants with increased reproductive effort. Although the transfer of nutrients from communities to plants has been demonstrated [e.g., 10], this is the first study, to our knowledge, that shows a relationship between pitcher plant phytotelma communities and components of plant fitness. Because we have simplified the characterization of communities in this study, it will be important to evaluate whether these patterns hold or are found to be stronger when more community members are included in measures of diversity.

## Acknowledgments

Marina Braine, Elsea Brown, Rob Evans, Steve Jaslow, Sara Melosh, Natrieifia Miller, Christian Muñoz Pineda, Kyle Pursel, Cathy Reimer, Sarah Seaton, Sophie Thompson, and Maggie

Winkler assisted with this research. Jen Rhode Ward, Caroline Kennedy, Mary Schultz, and an anonymous reviewer provided invaluable feedback on this manuscript. The findings and conclusions in this article are those of the authors and do not necessarily represent the views of the U.S. Fish and Wildlife Service.

## Author Contributions

**Conceptualization:** Rebecca E. Hale, Elise Powell, Leila Beikmohamadi, Mara L. Alexander.

**Data curation:** Rebecca E. Hale, Leila Beikmohamadi.

**Formal analysis:** Rebecca E. Hale, Mara L. Alexander.

**Funding acquisition:** Rebecca E. Hale, Elise Powell, Leila Beikmohamadi, Mara L. Alexander.

**Investigation:** Rebecca E. Hale, Elise Powell, Leila Beikmohamadi, Mara L. Alexander.

**Methodology:** Rebecca E. Hale, Elise Powell, Leila Beikmohamadi, Mara L. Alexander.

**Project administration:** Rebecca E. Hale.

**Resources:** Mara L. Alexander.

**Supervision:** Rebecca E. Hale, Mara L. Alexander.

**Writing – original draft:** Rebecca E. Hale, Elise Powell, Leila Beikmohamadi.

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
