## [Decision Letter · Decision Letter 0]

2 Jan 2020

PONE-D-19-33605

Symboint communities affect growth and reproductive effort of the purple pitcher plant (*Sarracenia purpurea* var. *montana*)

PLOS ONE

Dear Dr. Hale,

Thank you for submitting your manuscript to PLOS ONE. After careful consideration, we found that your MS provides valuable data from a well designed field study, but still requires a number of statistical improvements and a sharpening of the presentation of the manuscript. Therefore, we invite you to submit a revised version of the manuscript that addresses the points raised during the review process.

A number of potential referees with experience in this system were asked to review this MS, but only one referee agreed. Thus, I have also reviewed your paper for the same reason I agreed to act as AE: I began working with pitcher plant mosquitoes for my PhD (University of Rochester with Conrad Istock), and even visited some of these mountain bogs back in 1980 with help and advice from Dan Pittillo. I agree these are endangered habitats and thus represent a great opportunity to understand the community dynamics of these organisms.

The comments of the external referee are detailed and constructive. I too felt somewhat confused by the presentation of the statistical analyses and worried about the confidence in the regression results that seemed to be month-specific. With just two seasons of data, it is not surprising that such variation between months was found. I would have predicted  a larger effect of phenology given that these organisms overwinter in the pitchers. After thawing, I would think there could be similar community trajectories over summer that are dependent on rainfall. I see no discussion of local precipitation in your analyses.

I agree with the overall findings of the peer reviewer. The reviewer did apparently miss that you stated that your data have been made available in Dryad. The manuscript needs tightening with a clear and more direct connection between each of the statistical results and their implications for testing your original hypotheses. The data you have collected are valuable, and a better organized MS will help to attract readers to this study. On a minor point, please correct the spelling of symbiont: "Symboint" appears in the title and several other places in the text.

We would appreciate receiving your revised manuscript by Feb 16 2020 11:59PM. To enhance the reproducibility of your results, we recommend that if applicable you deposit your laboratory protocols in protocols.io, where a protocol can be assigned its own identifier (DOI) such that it can be cited independently in the future. For instructions see: http://journals.plos.org/plosone/s/submission-guidelines#loc-laboratory-protocols

We look forward to receiving your revised manuscript.

Kind regards,

William J. Etges

Academic Editor

PLOS ONE

Journal Requirements:

Reviewers' comments:

Reviewer's Responses to Questions

**Comments to the Author**

1. Is the manuscript technically sound, and do the data support the conclusions?

Reviewer #1: Partly

2. Has the statistical analysis been performed appropriately and rigorously? 

Reviewer #1: No

3. Have the authors made all data underlying the findings in their manuscript fully available?

Reviewer #1: No

4. Is the manuscript presented in an intelligible fashion and written in standard English?

Reviewer #1: No

5. Review Comments to the Author

Reviewer #1:

1. The study presents the results of original research.

2. The results reported have not been published elsewhere.

3. The experiments were performed adequately given that they represent field experiments; the statistics are unclear at best and in at least two instances, there is inconsistency between the significance claimed and the ratio of the parameter estimate and its standard error; Tables 4 & 6 appear to report the results of multiple regression, an approach that does not correct for fluid volume (a continuous variable) or site (a discrete variable). Full statistical tables as SI would have helped here. Due to large sample sizes, some significant correlations are low so that consistently reporting coefficients of determination or % reduction in total sum of squares would help the reader decide whether there was a substantive as well as statistically significant result.

4. I did not go through the discussion critically as the methodology was not clear, the results statistically equivocal, and the results went off on tangents from the hypotheses to be tested as stated in the introduction.

5. English was good, but the manuscript was muddled due to an indirect approach to the main hypotheses. The line numbers petered out towards the end of the manuscript.

6. The authors appeared to be very careful to (A) not reveal the coordinates of their localities due to the lamentable, but very real likelihood of poaching. They might give the total latitudinal and altitudinal range of their study to the nearest degree of longitude and 200m of elevation and (B) provide specific agency connections so other, legitimate researchers could seek permission to use their study sites.

7. Data reporting: I expect the raw data to be archived, but there was no obvious mention of plans to do so in the MS. Full statistical tables presented as supplemental information would have alleviated or increased my concern about statistical handling of the data.

Comments

I. Motivation for this study

As stated in the introduction, the motivation for this paper puts eight hypotheses (H’s in my analysis) at risk (lines 124-131):

Growth (= leaf production) increases with (H1) abundance and (H2) diversity of inquilines.

(H1) From top of Table 4, this hypothesis is supported in only one of three months in 2015 and not in any of the three months of 2016. For August, 2015, the reported P-value for Beta-1 is 0.005 but Beta-1 divided by its standard error is <1.96 and would appear not significant – some explanation in needed here.

(H2) From the bottom of Table 4, this hypothesis is supported by results from 2015 June and July, but not August and not by results from 2016. For July 2015, the reported P-value for Beta-1 is 0.021, but Beta-1 divided by its standard error is <<1.96 and would appear not significant - again, some explanation is needed here.

The middle of Table 4 has nothing to do with the stated hypotheses. If a significant effect of inquiline diversity were found, then testing for species richness or equitability would make sense for those comparisons where diversity did have an effect, but that is not the way the hypotheses are laid out or the results presented.

Reproductive effort (= number of flowers) increases with (H3) abundance and (H4) diversity of inquilines.

(H3) & (H4) From Table 6, top & bottom, respectively, neither hypothesis is supported in any month in either 2015 or 2016.

The middle of Table 6 has nothing to do with the stated hypotheses

The size of the inquiline population (= number of inquilines in a leaf) (H5) increases with host population size and (H6) decreases with distance to nearest neighbor (= site). The pertinent results are provided without table or illustration in the first paragraph of “Population size and isolation” (The lines are not numbered in this section). Neither hypothesis is supported.

The diversity of the inquiline population (H7) increases with host population size and (H8) decreases with distance to nearest neighbor. The pertinent results are provided without table or illustration in the second paragraph of “Population size and isolation.” H7 is supported; H8 is not.

What is needed here is a clear path to the tests of the staged hypotheses and then a discussion of what the results of those results mean, uncluttered with tangential material that reflects mainly ad hoc “what if’s.” The authors make a good case for pursuing the main hypotheses in the introduction and then muddle their results with other information that is incidental to the stated hypotheses to be tested, making for tedious reading and distracting from their primary mission.

II. Some other comments

1. The authors need to define rosettes, clumps and population sizes. Individual pitcher plants are comprised of one or more meristems – does a rosette correspond to a meristem or a plant that is composed of one or more meristems? Are clumps individual plants composed of one or more meristems or groups of plants and meristems that are too dense to separate into their component parts without damaging the plants (this is a genuine, practical concern, but if that is their concern, they need to state it in their definitions). They define population size by number of “the number of distinct clumps of rosettes. Although clumps may include multiple, genetically distinct plants, we assume that the number of clumps is correlated with the number of genetically distinct plants… ” (lines 140-142). So the appropriate within site-month-year independent unit is “clump,” even though census data between months or years within a clump are not independent, normalized clump data are more independent that rosettes within a clump (Lines 253-254) or leaves within a rosette. One meristem puts out numerous leaves but can produce zero, one or, rarely, two flowers; similarly, one plant with multiple meristems can produce zero, one, or multiple flowers. It all comes down to what is a replicate unit within a site-month-year. Multiple leaves on the same plant are not independent and meristems/plants (rosettes?) in a clump are not independent – so maybe the unit to be entered into the error term of their analyses are number of clumps, not number rosettes within clumps, and not leaves within rosettes. Lines 244-248 & 259-265 suggest that they are using individual leaves as independent estimates for the error term in their F-tests and are therefore prone to Type I error.

2. Line 158: What randomization procedure was used?

3. Lines 181-189, 203: This study concerns numbers and diversity of arthropods exclusively. Since rotifers and unicellular eukaryotes have been also been quantified in previous studies, the reason for their exclusion should be stated. I suspect the reason is practical and within the time-frame and knowledge of student projects – and these are very real considerations. Nonetheless, at the very least, the title of the paper should be “Arthropod” or “Arthropod symbiont” instead of “Symbiont” communities.

4. Line 212. Why weren’t interaction terms included? Certainly R has the facility to do so.

5. Lines 206-217. Here is where the interaction terms become informative. In lines 215-216, distinguishing individual rosettes (within clumps?) proved a problem and caused them to omit the site with the smallest size (Table 1; does “size” refer to pitchers, rosettes, or clumps?). Looking “for consistency across months” does not substitute for a multiple comparisons test, especially if “months within years and sites” is not specifically entered into the model. How did they quantify “look for?”

I am not saying that analysis of their data set is straightforward, but their approach for testing their stated hypotheses with confounding years and months is unclear and the error terms in t- and F-tests appear based on number of leaves, which are not independent observations within rosettes or clumps.

I like their rationale for pursuing this study and their stated hypotheses are clear; it is making a similarly clear path to the testing of these hypotheses that makes their results equivocal. I suggest that they define their terms, break down their hypotheses as I have done, determine what is the minimum information needed to resolve those hypotheses, justify the statistical approach (stating explicitly the constraints imposed by field conditions – which are what provide substance to their study and stating explicitly what units – leaves, rosettes, clumps, sites – were used to generate the error term), and forget all ancillary observations, data, analyses, and discussion. Their concluding statement will be dicey, as their results do not support the global conclusion “that phytotelma [sic] communities benefit host pitcher plants with increased vegetative growth and reproductive effort.” I don’t doubt that this conclusion may be true in nature, but (1) the data in Table 4 do not support the conclusion of abundance of inquilines increasing growth and may support the conclusion of inquiline diversity increasing growth in some months but not others and (2) the data in Table 6 do not support the conclusion that inquiline abundance or diversity increases reproductive effort.

6. PLOS authors have the option to publish the peer review history of their article (what does this mean?). If published, this will include your full peer review and any attached files.

Reviewer #1: No

---

## [Author Response · Author response to Decision Letter 0]

27 Feb 2020

Please see the attached letter in which I respond to all reviewer comments.

---

## [Decision Letter · Decision Letter 1]

6 Mar 2020

PONE-D-19-33605R1

Effects of arthropod inquilines on growth and reproductive effort among metacommunities of the purple pitcher plant (*Sarracenia purpurea* var. *montana*)

PLOS ONE

Dear Dr. Hale,

Thank you for submitting your manuscript to PLOS ONE. After careful consideration, we feel that it has merit but could be improved substantially prior to possible publication. Therefore, we invite you to submit a revised version of the manuscript that addresses the points raised during the review process.

I commend the authors for performing a serious revision of their manuscript and responding to the many comments and criticisms from the peer reviewer and me. The peer reviewer has responded in an attached pdf with some cogent comments concerning the presentation of your results in this manuscript. I think the suggestions made here are appropriate and would crystalize the impact of the paper for readers of this journal. While field studies over several seasons are inherently valuable, there are times when natural variability makes significant biological patterns difficult to discern. This takes nothing away from the data set presented here, but more focus on the statistically relevant and significant results will improve the paper.

We would appreciate receiving your revised manuscript by Apr 20 2020 11:59PM. To enhance the reproducibility of your results, we recommend that if applicable you deposit your laboratory protocols in protocols.io, where a protocol can be assigned its own identifier (DOI) such that it can be cited independently in the future. For instructions see: http://journals.plos.org/plosone/s/submission-guidelines#loc-laboratory-protocols

We look forward to receiving your revised manuscript.

Kind regards,

William J. Etges

Academic Editor

PLOS ONE

Reviewers' comments:

Reviewer's Responses to Questions

**Comments to the Author**

1. If the authors have adequately addressed your comments raised in a previous round of review and you feel that this manuscript is now acceptable for publication, you may indicate that here to bypass the “Comments to the Author” section, enter your conflict of interest statement in the “Confidential to Editor” section, and submit your "Accept" recommendation.

Reviewer #1: (No Response)

2. Is the manuscript technically sound, and do the data support the conclusions?

Reviewer #1: Partly

3. Has the statistical analysis been performed appropriately and rigorously? 

Reviewer #1: Yes

4. Have the authors made all data underlying the findings in their manuscript fully available?

Reviewer #1: Yes

5. Is the manuscript presented in an intelligible fashion and written in standard English?

Reviewer #1: Yes

6. Review Comments to the Author

Reviewer #1: See attached document.

7. PLOS authors have the option to publish the peer review history of their article (what does this mean?). If published, this will include your full peer review and any attached files.

Reviewer #1: No

---

## [Author Response · Author response to Decision Letter 1]

17 Apr 2020

We have responded to reviewer comments in the uploaded cover letter.

---

## [Editor Report · Decision Letter 2]

23 Apr 2020

Effects of arthropod inquilines on growth and reproductive effort among metacommunities of the purple pitcher plant (*Sarracenia purpurea* var. *montana*)

PONE-D-19-33605R2

Dear Dr. Hale,

We are pleased to inform you that your manuscript has been judged scientifically suitable for publication and will be formally accepted for publication once it complies with all outstanding technical requirements.

There is one additional statement that you ought to add - please include reviewer 1 in your acknowledgements section. I think these comments really improved your presentation.

With kind regards,

William J. Etges

Academic Editor

PLOS ONE
---

## [Editor Report · Acceptance letter]

28 Apr 2020

PONE-D-19-33605R2 

Effects of arthropod inquilines on growth and reproductive effort among metacommunities of the purple pitcher plant (*Sarracenia purpurea* var. *montana*) 

Dear Dr. Hale:

I am pleased to inform you that your manuscript has been deemed suitable for publication in PLOS ONE. Congratulations! Your manuscript is now with our production department. 

With kind regards,

on behalf of

Dr. William J. Etges 

Academic Editor

PLOS ONE